# OpenReview forum: "Large Language Model Unlearning via Embedding-Corrupted Prompts"
_NeurIPS.cc/2024/Conference — NeurIPS 2024 poster_

### Official Review · Reviewer_qM3a · 2024-07-10

**Soundness:** 3
**Presentation:** 3
**Contribution:** 3
**Rating:** 6
**Confidence:** 5

**Summary:**

This paper introduces a novel approach named Embedding-COrrupted (ECO) Prompts for efficient unlearning in LLMs. ECO maintains an unlearning state during inference by utilizing a prompt classifier, eliminating the need to modify LLMs. This classifier identifies, corrupts, and safeguards prompts to be forgotten in the embedding space during inference. Extensive experiments show that ECO Prompts achieve effective unlearning with minimal side effects among various LLMs on several unlearning benchmarks.

**Strengths:**

1. The authors propose a new method for unlearning during inference directly, eliminating the need to update LLMs. This approach somehow addresses the challenges and costs of re-training or fine-tuning LLMs for unlearning purposes. It makes some contributions to this particular field of research.
2. The ECO Prompts method is applicable to a wide range of knowledge entanglement and unlearning tasks, demonstrating strong generalizability across various LLMs. Additionally, it shows potential for integration with other unlearning techniques in the future.
3. The paper is clearly written and well-organized. It is easy to follow the authors' ideas and understand their approaches. The authors use clear figure, i.e., Figure 1, to show the procedure of their method. The notations and experimental results are clear and easy to read.
4. The authors have done extensive experiments to demonstrate the effectiveness of ECO Prompts using various LLMs across several unlearning benchmarks.
5. The authors have provided a comprehensive literature review and show the advantages of ECO Prompts compared to several existing related works.

**Weaknesses:**

1. Although using a prompt classifier during inference to identify the unlearning state is straightforward, it also makes it less applicable for end users. The pre-trained prompt classifier requires additional time and effort for unlearning tasks. We need additional datasets to train this prompt classifier. Additionally, it raises some concerns regarding trustworthiness and safety during application.
2. Another issue is that the proposed approach does not truly help LLMs unlearn or forget certain knowledge. Some potential prompts may still trigger harmful responses or copyrighted content. The empirically similar performance over the metrics set in Line 117 does not really mean unlearning.
3. There are some concepts that need further clarification or justification. For instance, what do the labels, i.e., [0,0,1,1,0], in Figure 1 mean? What is the embedding function $E$, and how can we detach it from a black-box LLM?  The claim about ``potential fuzzy boundary between retaining and forgetting" needs further justification or citations. The $\textbf{Note}$ in Section 3.3 is not quite clear.

**Questions:**

1. What does $r$ mean in Eq. (2)?
2. How do you design the dataset to train the prompt classifier?
3. How do you get the surrogate retain metric value $\hat{v}_r$?
4. Does it mean to use the same $\sigma$ in Line 237?

**Limitations:**

1. The performance of ECO highly depends on the performance of the prompt classifier.
2. The performance of ECO could be affected by potential attacks against the prompts.

---

> ### Author Rebuttal · Authors · 2024-08-06
>
> We appreciate your thorough feedback and constructive criticism of our paper. Below, we respond to the weaknesses and questions you raised in your review.
>
> > Although using a prompt classifier during inference to identify the unlearning state is straightforward, it also makes it less applicable for end users.
>
> As we mentioned throughout the paper, our proposed method targets large and powerful models with chat interface or API access. This is because gradient-based methods are usually extremely expensive to perform, especially in scenarios where frequent unlearning requests are the norm, which is the main focus in this work.
>
> > The pre-trained prompt classifier requires additional time and effort for unlearning tasks. We need additional datasets to train this prompt classifier.
>
> For most unlearning scenarios, we usually need at least a forget set to unlearn, just like the unlearning baselines we have compared to. A retain set can help mitigate performance degradation, and can be easy to obtain. For example, SOTA unlearning methods like SCRUB (https://arxiv.org/abs/2302.09880), LLMU (https://arxiv.org/abs/2310.10683), RMU (https://arxiv.org/abs/2403.03218) require both retain data and forget data to work well, which we also use to train our classifiers. So we are using the same data as one would use in a traditional unlearning algorithm, and no additional time and effort is required.
>
> In fact, training a classifier is way cheaper than updating the LLM itself. For all classifiers we have trained, the most “expensive” one only requires 30 minutes on a single NVIDIA RTX 4090, while a decent unlearning method like SCRUB takes more than 12 hours on two NVIDIA A100’s.
>
> > Some potential prompts may still trigger harmful responses or copyrighted content.
>
> We conducted an additional study using multiple types of challenging queries either written by humans, including rephrasing, adversarial, adding irrelevant context, jailbreak, and including only keywords. Below are the false positive/negative rates (in percentage) of the classifier on the perturbed queries:
>
> |Perturbation type|False Positive Rate (%)|False Negative Rate (%)|
> |--------------------------|-----------------------|-----------------------|
> |None|0.0|0.0|
> |Rephrased|0.14|1.5|
> |Adversarial|0.2|1.5|
> |Withirrelevantcontext|0.17|0.5|
> |Withjailbreak-likesyntax|1.65|7.52|
> |Keywordsandshortphrases|0.28|2.51|
>
> Based on these statistics, we believe that our classifiers remain robust under these common perturbations, even without training on these types of data. We continue to explore if we could improve the robustness of our classifiers further. We construct another set of perturbed prompts, all synthetically generated by Llama-3-70B-Instruct, and have no overlap with the previous perturbed evaluation set, and has no knowledge about the perturbation types. We use this set of prompts to train our classifier, and observe that both the false positive rate and the false negative rate can be further improved.
>
> |Perturbation type|False Positive Rate (%)|False Negative Rate (%)|
> |--------------------------|-----------------------|-----------------------|
> |None|0.0|0.0|
> |Rephrased|0.03|0.5|
> |Adversarial|0.06|0.75|
> |With irrelevant context|0.0|0.25|
> |With jailbreak-like syntax|0.08|1.75|
> |Keywords/short phrases|0.0|0.5|
>
> We also perform additional training on the WMDP and copyrighted content classifiers and the result still holds:
>
> WMDP
>
> |Perturbation type|False Positive Rate (%)|False Negative Rate (%)|
> |--------------------------|-----------------------|-----------------------|
> |None|0.00|0.00|
> |Rephrased|0.01|1.00|
> |Adversarial|0.1|1.1|
> |With relevant context|0.00|0.83|
> |With jailbreak-like syntax|0.07|3.50|
> |Keywords/short phrases|0.00|1.03|
>
> Copyrighted content
>
> |Perturbation type|False Positive Rate (%)|False Negative Rate (%)|
> |--------------------------|-----------------------|-----------------------|
> |None|0.0|0.0|
> |Rephrased|0.01|0.2|
> |Adversarial|0.03|1.81|
> |With relevant context|0.00|0.37|
> |With jailbreak-like syntax|0.09|2.63|
> |Keywords/short phrases|0.0|0.18|
>
> > How do you get the surrogate retain metric value $\hat{v}_{r}$?
>
> Equation (8) aims to optimize the retained model's performance on the relevant task. However, since obtaining a retained model is impractical, we use a surrogate score to estimate its performance. The $\hat{v}_r$ in Equation (8) represents an estimated score for the retained model's performance on evaluation metrics. For example, in the WMDP task, we set $\hat{v}_r = 0.25$ to match the random-guessing level (25%) for multiple-choice tasks. For more complex metrics in the TOFU and copyrighted content tasks, $\hat{v}_r$ is determined using human-written responses for 100 training prompts, simulating the expected retained model responses (e.g., refusals, "I don't know," incorrect answers). These pseudo responses are used to calculate the surrogate retain metric value.
>
> > How do you design the dataset to train the prompt classifier?
>
> For the TOFU dataset, both the retain and forget sets are available from the dataset itself, so we can directly use them. We follow the practice of the TOFU paper, which uses both the full retain set and the forget set for unlearning.
>
> For the WMDP dataset, we use GPT-4-Turbo to generate 100 synthetic questions for each of the biology, chemistry, and cybersecurity as the training data for the classifier. We also validate that none of the generated questions appear in the forget set questions.
>
> For the copyrighted content classifiers, we use the actual copyrighted content as the positive examples (which we hope the classifiers can predict positive). We use books and news articles from different sources as negative samples for the Book and News tasks, respectively.
>
> We provide further detail and clarification on how the training sets of the classifiers are constructed in Appendix C.3.

---

> ### Author Response · Authors · 2024-08-06
> **Additional Comment to Reviewer Response**
>
> > There are some concepts that need further clarification or justification. For instance, what do the labels, i.e., [0,0,1,1,0], in Figure 1 mean?
>
> The [0, 0, 1, 1, 0] in Figure 1 is a corruption mask, where 0's mean the token embeddings at those positions are not corrupted, and 1's mean the token embeddings are corrupted at those positions. On TOFU, we use a separate token classifier to select tokens that are relevant to names and only corrupt those tokens' embeddings. On WMDP and copyrighted content tasks, we corrupt all tokens.
>
> > The **Note** in Section 3.3 is not quite clear.
>
> There might also be cases where the surrogate retain value is not possible to obtain (e.g., no suitable evaluation metrics exist, or the chosen evaluation metric might not be enough to reflect if a model actually exhibits forgetting in the real-world use cases, etc.). In these cases, the model service provider could instead conduct red teaming with human annotators using responses generated by different $\sigma$'s.
>
> > What is the embedding function , and how can we detach it from a black-box LLM?
>
> The embedding function is the embedding layer in an LLM. We separate it from the model because the proposed corruption scheme only applies to the output of the embedding layer and does not affect all other layers in the model. Specifically, we consider an LLM as a function $f=\tilde{h}\circ\mathbf{e}(x)$, where $\mathbf{e}(x)$ produces the token embeddings, and $\tilde{h}$ is mapping from the token embeddings to the last layer output logits.
>
> > The claim about ``potential fuzzy boundary between retaining and forgetting" needs further justification or citations.
>
> We will add citations to our reference to the "potential fuzzy boundary between retaining and forgetting" with [1, 2, 3], where they contain evidence showing unlearning in one domain could induce harm in closely-related or irrelevant domains.
>
> > What does $r$ mean in Eq. (2)?
>
> The $r$ and $f$ in Equation (2) mean retain and forget, respectively. Here, we mean to say that if the classifier's probability of the prompt being in the forget distribution is larger than the probability of the prompt being in the retail distribution.
>
> > Does it mean to use the same $\sigma$ in Line 237?
>
> Yes, this means that we run the optimization once on the task of forgetting 1% examples in the TOFU dataset and obtain a $\sigma^*$. Then, we reuse the same $\sigma^*$ (obtained on the 1%) in the 5% and 10% setting for both Llama 2 and Phi-1.5.
>
> [1] Maini, P., Feng, Z., Schwarzschild, A., Lipton, Z. C., & Kolter, J. Z. (2024). Tofu: A task of fictitious unlearning for llms. arXiv preprint arXiv:2401.06121.
> [2] Lynch, A., Guo, P., Ewart, A., Casper, S., & Hadfield-Menell, D. (2024). Eight methods to evaluate robust unlearning in llms. arXiv preprint arXiv:2402.16835.
> [3] Qiu, X., Shen, W. F., Chen, Y., Cancedda, N., Stenetorp, P., & Lane, N. D. (2024). PISTOL: Dataset Compilation Pipeline for Structural Unlearning of LLMs. arXiv preprint arXiv:2406.16810.

---

> > ### Comment · Reviewer_qM3a · 2024-08-12
> > **Thank you for your responses.**
> >
> > I have read the authors' responses. Most of my concerns have been addressed. I will keep my score as 6 Weak Accept.

---

> > > ### Author Response · Authors · 2024-08-12
> > > **Thank You**
> > >
> > > We are pleased to have addressed most of your concerns. Thank you again for your thoughtful feedback and questions in the reivew, which will help us in refining our paper during the revision process.

---

### Official Review · Reviewer_UAgG · 2024-07-11

**Soundness:** 3
**Presentation:** 3
**Contribution:** 3
**Rating:** 5
**Confidence:** 4

**Summary:**

This paper proposes a straightforward but well-crafted and effective method to implement unlearning for LLMs through an inference-time intervention. The method first trains a classifier, to determine whether a prompt contains material to be forgotten. Then, it corrupts the prompt to the LLM so that its response won't reflect the scrubbed topic. Extensive experiments on realistic tasks show that information can be reliably forgotten with minimal reduction in model efficacy on other topics.

**Strengths:**

- Although the basic idea is straightforward, the design choices seem to be well thought-out. For instance, thresholding the classifier with conformal prediction, or the zeroth order optimizer to learn flexible corruptions while only having API access.
- The three tasks (entity unlearning, hazardous material, and copyright content) were well chosen and interesting and the metrics for each were fitting
- Extensive experiments and reference list
- Well-written intro and motivation. For instance, the authors point out that inference time interventions are much more scalable than gradient-based approaches as we go to large models.

**Weaknesses:**

Using classifiers as a guard rail is a simple and longstanding approach in practice. One of the main innovations of this paper is the corrupting mechanism, to return results similar to the "retained" model rather than a simple template. However, most (all?) of the results in the main text would seem to be equally well served by the simple classifier + template mechanism. Can you clarify where the benefit of the corrupted prompt responses is shown? You talk about the expected metric ratio of unlearned versus retained under the forget prompts as the measure that would reflect this, but I didn't see any results like this. (But I didn't have time to look at all appendix experiments. This is a central point of the paper, though, it should be highlighted in the main text.)

**Questions:**

My main question is about demonstrating the value of the corruption mechanism, and I would modify my score based on a better understanding of this.

Minor comments / questions:
- Lines 100-104 were very confusing. I think maybe a notation change was not reflected. I eventually understood what was meant from context.  (also match changes to footnote 2)
- Line 79-80 were confusing. You assume in-distribution queries, and I think you meant to say users *do not* attempt to jailbreak...
- 197 "we only corrupt the first dimension of each token's embedding". That really surprised me, but I guess I'll have to check out the related work to understand.
- It seems like the surrogate model plays an important role. I didn't understand it from the main text.
- Were there any cases where you could include the classifier + template response in the results? It would be interesting to see how much better the metrics match from doing corruption instead of returning a template

**Limitations:**

Limitations were discussed in the appendix. (e.g. that the method could be subverted by adversarial attacks.)
Harmful applications were also discussed (e.g. a service provider could conceal information from users in a relatively hard-to-detect way.)

---

> ### Author Rebuttal · Authors · 2024-08-06
>
> Thank you for your exhaustive feedback and the constructive criticism of our paper. Below we address the weaknesses and questions provided in the review above.
>
> > However, most (all?) of the results in the main text would seem to be equally well served by the simple classifier + template mechanism. Can you clarify where the benefit of the corrupted prompt responses is shown?
>
> Indeed, a simple classifier and a template mechanism can already mitigate many risk types. The benefit of the corrupted prompt responses, as opposed to a simple classifier + template mechanism, lies in ensuring indistinguishability, thereby mitigating privacy risks. In our early experiments, we find that even if a model has not been trained on a piece of data, it frequently provides a hallucinated response instead of answering “I don’t know.” Therefore, our aim is for the corruption mechanism to generate results akin to the "retained" model, effectively masking whether specific data has been unlearned.
>
> In contrast, a template response could inadvertently reveal the classifier's training on a particular individual. For example, in the context of entity unlearning, when the model generates a template response, an attacker would know that it’s highly likely the classifier has been trained to identify this individual. Having this knowledge, an attacker could continue to exploit the individual’s characteristics, behaviors, or conditions.
>
> > You talk about the expected metric ratio of unlearned versus retained under the forget prompts as the measure that would reflect this, but I didn't see any results like this.
>
> For the expected metric ratio of unlearned versus retained, the ratio is reflected in all three groups of experiments:
> For TOFU, it is the forget quality shown in Figure 2. This is the distributional similarity between the outputs of an unlearned model and a retained model, which is even a stronger guarantee. A larger forget quality indicates a ratio close to 1, in distribution.
> The WMDP authors assume that a model that has not been trained on relevant domain knowledge should have random-guessing accuracy on the multiple-choice questions. The closer the score is to 0.25, the closer the expected ratio of unlearned to retained metrics is to 1.
>
> For copyrighted content unlearning, we use average similarity gap (ASG), which measures the gap (retained model outputs vs original text similarity) - (unlearn model outputs vs original text similarity) shown in Table 3. The closer the gap, the closer the expected ratio of unlearned to retained metrics is to 1.
>
> > It seems like the surrogate model plays an important role. I didn't understand it from the main text.
>
> Here, while we use the term surrogate model, the surrogate retain metric value does not come from a separate model. Instead, the $\hat{v}_r$ in Equation (8) represents a score one "thinks" a retained model would obtain on the relevant evaluation metrics. Given that the retained model is not accessible, we have to derive this score based on the problem. For example, in the WMDP task, the goal of unlearning is to reduce the accuracy of relevant multiple choice tasks to a random-guessing level (25% because all questions have four possible choices), which we expect a retained model would have. So we can set the surrogate retain metric value $\hat{v}_r = 0.25$ directly, and running the optimization will then push the "unlearned metric value" closer to the surrogate retain metric value $\hat{v}_r$.
>
> > Were there any cases where you could include the classifier + template response in the results? It would be interesting to see how much better the metrics match from doing corruption instead of returning a template
>
> For TOFU, the core evaluation metric, forget quality, examines if the unlearned model's raw output distribution matches that of the retained model. This prevents us from doing a classifier + template response, because the evaluation does not rely on the text output. However, as ROUGE-L is one of the evaluation metrics used in TOFU and it is a text-based similarity measure, we compare if the ROUGE-L of the retained model is close to the template responses using the same metric in Table 3. Below we show that ECO maintains the highest similarity to the retained model’s responses.
>
> |Model/Method|ROUGE-L|
> |--------------------------------|-------|
> |Original Model Before Unlearning|0.9738|
> |Retainedmodel|0.3951|
> |ECO|0.3067|
> |Templates|0.0210|
>
> For copyrighted content unlearning, we also include the results from using template response (on BBC News task) as follows. The following table corresponds to Table 3.
>
> |Method|ASG(↓)|
> |-----------|---------|
> |Original|71.2|
> |Retain|0|
> |Fine-tune|48.5|
> |GA|12.4|
> |GD|26.3|
> |KL|4.1|
> |Mismatch|3.9|
> |SCRUB|14.3|
> |LLMU|11.9|
> |ECO(Ours)|1.6|
> |Template|11.3|
>
> > Line 79-80 were confusing. You assume in-distribution queries, and I think you meant to say users _do not_ attempt to jailbreak...
>
> Yes, we meant to say that the users do not attempt to jailbreak the classifiers. We also conducted additional experiments, which show that our classifiers are also robust against different types of perturbed queries even if we did not explicitly train the classifiers to be robust. We include the detailed response in the comment below.

---

> > ### Author Response · Authors · 2024-08-08
> > **Additional experiments on robustness of classifiers**
> >
> > We conducted an additional study using multiple types of challenging queries either written by humans, including rephrasing, adversarial, adding irrelevant context, jailbreak, and including only keywords. Below are the false positive/negative rates (in percentage) of the classifier on the perturbed queries:
> >
> > |Perturbation type|False Positive Rate (%)|False Negative Rate (%)|
> > |--------------------------|-----------------------|-----------------------|
> > |None|0.0|0.0|
> > |Rephrased|0.14|1.5|
> > |Adversarial|0.2|1.5|
> > |Withirrelevantcontext|0.17|0.5|
> > |Withjailbreak-likesyntax|1.65|7.52|
> > |Keywordsandshortphrases|0.28|2.51|
> >
> > Based on these statistics, we believe that our classifiers remain robust under these common perturbations, even without training on these types of data. We continue to explore if we could improve the robustness of our classifiers further. We construct another set of perturbed prompts, all synthetically generated by Llama-3-70B-Instruct, and have no overlap with the previous perturbed evaluation set, and has no knowledge about the perturbation types. We use this set of prompts to train our classifier, and observe that both the false positive rate and the false negative rate can be further improved.
> >
> > |Perturbation type|False Positive Rate (%)|False Negative Rate (%)|
> > |--------------------------|-----------------------|-----------------------|
> > |None|0.0|0.0|
> > |Rephrased|0.03|0.5|
> > |Adversarial|0.06|0.75|
> > |With irrelevant context|0.0|0.25|
> > |With jailbreak-like syntax|0.08|1.75|
> > |Keywords/short phrases|0.0|0.5|
> >
> > We also perform additional training on the WMDP and copyrighted content classifiers and the result still holds:
> >
> > WMDP
> >
> > |Perturbation type|False Positive Rate (%)|False Negative Rate (%)|
> > |--------------------------|-----------------------|-----------------------|
> > |None|0.00|0.00|
> > |Rephrased|0.01|1.00|
> > |Adversarial|0.1|1.1|
> > |With relevant context|0.00|0.83|
> > |With jailbreak-like syntax|0.07|3.50|
> > |Keywords/short phrases|0.00|1.03|
> >
> > Copyrighted content
> >
> > |Perturbation type|False Positive Rate (%)|False Negative Rate (%)|
> > |--------------------------|-----------------------|-----------------------|
> > |None|0.0|0.0|
> > |Rephrased|0.01|0.2|
> > |Adversarial|0.03|1.81|
> > |With relevant context|0.00|0.37|
> > |With jailbreak-like syntax|0.09|2.63|
> > |Keywords/short phrases|0.0|0.18|

---

> > ### Comment · Reviewer_UAgG · 2024-08-09
> > **Response**
> >
> > # Weaker points
> > - I disagree about the "forget quality" metric. You say it's a stronger distributional matching metric, but this is on multiple choice, not on raw text. You can use the classifier (without the corruption) and then return equal probability to all multiple choice entries (instead of a template, which I agree isn't relevant here). I believe this would be equally efficacious for your forget quality metrics, and it does not use the main result of your paper (how to add corruption).
> >
> > - Your points about privacy and hallucination also make sense. It would strengthen the paper to have experiments that probe these more directly.
> >
> > # Stronger points
> > - I agree the ASG result looks like a good example where the corruption definitely improves over just a classifier.
> >
> > - The ROUGE score for TOFU outputs also makes sense to me as a metric to help verify this. Was it in the original paper and I missed it? If not, I recommend adding it.
> >
> > I think you've demonstrated some benefits that accrue to the corruption approach, and will raise my score appropriately.

---

> ### Author Response · Authors · 2024-08-09
> **Clarification on Forget Quality and Additiona Experiments**
>
> We appreciate your kind recognition of our efforts and your thoughtful feedback. Below, we would like to provide clarification regarding the forget quality metric and additional questions in the comment above.
>
> 1. **The forget quality metric is not based on multiple choice but on the output distribution of the retained model and the unlearned model**. The TOFU paper [1] uses the forget quality to measure indistinguishability in the output distribution of two models (retained and unlearned), given some data (which we explain in the following paragraph). To compute the forget quality, we first need to compute another metric known as the truth ratio (see section 2.2.2 in https://arxiv.org/pdf/2401.06121). The truth ratio measures, given a model and a prompt, how likely the correct answer is to an incorrect answer. Below, we quote the original interpretation from the TOFU paper for a better understanding:
>
> > This ratio informs us of the degree to which the unlearning algorithm removed the information to be forgotten. Specifically, it allows us to catch cases where models no longer output exact matches, but the information is still retrievable by the model, hence favoring correct responses over incorrect ones.
>
> To further compute the forget quality, we perform a Kolmogorov–Smirnov (KS) test, a statistical test measuring the difference between two cumulative distributions. We perform KS test on the two distributions of truth ratio calculated from the retain model and the unlearned model, respectively, following exactly the method in the TOFU paper. The KS test results in a p-value, and a high p-value means that we cannot reject the null hypothesis that the two distributions are the same. In our experiments, we obtained p-values over 0.95 in all forget sizes and models considered. Therefore, our approach achieved distribution similarity to a retained model in the models' output probability distribution.
>
> Note that the distributional similarity in the output distribution is a direct consequence of making the model answering the corrupted prompt. Another way to put it is: the model's probability of generating the original answer will be reduced significantly on the corrupted tokens.
>
> 2. We do provide **a full evaluation of text-based metric, ROUGE-L** (between the generated text after unlearning and the original correct answer), in Table 10 and Table 11 in the appendix (page 30 and 31), due to the page constraint. Our method can have ROUGE-L scores close to that of the retained model, and can significantly reduce the ROUGE-L score to extremely low (if we want to). We also provide **real generated examples on page 32**, which again shows the effect of unlearning on text.
>
> 3. On WMDP, we provided **a probing experiment** in D.3.1 (page 35). While this task is on multiple-choice, we show that attackers cannot recover the correct answer even if one gains access to the raw model logits.
>
> We hope the above clarification answers your question in the comment above.
>
> [1] Maini, P., Feng, Z., Schwarzschild, A., Lipton, Z. C., & Kolter, J. Z. (2024). Tofu: A task of fictitious unlearning for llms. arXiv preprint arXiv:2401.06121.

---

> > ### Comment · Reviewer_UAgG · 2024-08-09
> > **Forget quality clarification**
> >
> > Thanks for the clarification, the paper link makes it more clear. But you could imagine a classifier only model (without corruption) that says that, if the erased concept appears according the classifier, use a trivial LM that says all answers are equally likely. This would likely differ from the distribution given by the retain model, though, and maybe not get as good of a forget quality as your corruption approach.

---

### Official Review · Reviewer_jeTk · 2024-07-13

**Soundness:** 2
**Presentation:** 3
**Contribution:** 2
**Rating:** 5
**Confidence:** 3

**Summary:**

This paper proposes a lightweight method for unlearning in large language models (LLMs), based on corrupting the embedding of a query related to the forget data identified by an external classifier during LLM inference. The experimental results demonstrate the effectiveness of this method on various benchmarks, such as ToFU/WMDP, and across multiple LLMs.

**Strengths:**

* The proposed method is very simple and lightweight.
* The experimental results show strong performance compared to baseline methods.

**Weaknesses:**

* The optimization objective in Equations (8) and (9) for the corruption function incorporates the distance measure function between the corrupted LLM and the retained LLM, which is saying some 'unlearned metric value'. This raises concerns for me. What specific measure function was employed in the experiments? If the same metric function used for evaluation is employed, it could offer an unfair advantage over other baseline methods. I tried to find this detail in the paper but failed. Given its importance, I would like to initially provide a negative score. Please correct me if I have missed it in the paper.

**Questions:**

* I am confused about the selection of the surrogate retain metric value in line 186, can you elaborate more? What do you mean by if it's not possible to obtain?
* How many token embeddings are selected to be corrupted? Figure 1 only shows partial corruption of the input, while Equations (10) and (11) indicate corruption of the entire input embedding e, which is very confusing.
* The false positive rate of the employed classifier could inevitably affect overall performance. It's possible the classifier might exhibit false positives on challenging examples, such as perturbed queries related to the knowledge intended to be forgotten. Since the ToFU dataset contains perturbed versions of forget and retain queries, I am curious about the classifier's performance on this subset of data, which might yield a high false positive rate.
* Given that the forget data for training is quite small, for example, 400 samples for ToFU-10%, is there a risk that the classifier overfits on the training forget data? Can it generalize to other paraphrased versions of the forget data?

**Limitations:**

Authors discussed limitation and societal impact.

---

> ### Author Rebuttal · Authors · 2024-08-06
>
> We are grateful for your comprehensive review and the valuable insights you provided. Below we address the weaknesses and questions provided in the review above.
>
> > The optimization objective in Equations (8) and (9) for the corruption function incorporates the distance measure function between the corrupted LLM and the retained LLM, which is saying some 'unlearned metric value'. This raises concerns for me. What specific measure function was employed in the experiments?
>
> Equation (8) aims to optimize the retained model's performance on the relevant task. However, since obtaining a retained model is impractical, we use a surrogate score to estimate its performance. The $\hat{v}_r$ in Equation (8) represents an estimated score for the retained model's performance on evaluation metrics. For example, in the WMDP task, we set $\hat{v}_r = 0.25$ to match the random-guessing level (25%) for multiple-choice tasks. For more complex metrics in the TOFU and copyrighted content tasks, $\hat{v}_r$ is determined using human-written responses for 100 training prompts, simulating the expected retained model responses (e.g., refusals, "I don't know," incorrect answers). These pseudo responses are used to calculate the surrogate retain metric value.
>
> We would also like to highlight that our method imposes an unlearned state over the LLM subjected to unlearning. This means that an LLM service provider could simply adjust the corruption strength value $\sigma$ in real time. The optimal strength obtained from the optimization stage can serve as an initial guess, and one can directly increase $\sigma$ for more unlearning or disable it to recover the original LLM based on the specific unlearning request during model serving. Compared to other baseline methods requiring offline model updates, having an online tunable $\sigma$ which imposes an unlearning state is a part of the contribution of our approach and a clear advantage over those baselines.
>
> > I am confused about the selection of the surrogate retain metric value in line 186, can you elaborate more? What do you mean by if it's not possible to obtain?
>
> There might also be cases where the surrogate retain value is not possible to obtain in real-world (e.g., no suitable evaluation metrics exist, or the chosen evaluation metric might not be enough to reflect if a model actually exhibits forgetting in the real-world use cases, etc.). In these cases, the model service provider could instead conduct red teaming with human annotators using responses generated by different $\sigma$'s, and select the optimal $\sigma^*$ based on human feedback.
>
> > How many token embeddings are selected to be corrupted? Figure 1 only shows partial corruption of the input, while Equations (10) and (11) indicate corruption of the entire input embedding e, which is very confusing.
>
> On the TOFU dataset, we also use an additional token classifier to select only name tokens to corrupt, as this task is about forgetting entities. For WMDP and copyrighted content removal, we corrupt all tokens in the prompt.
>
> > The false positive rate of the employed classifier could inevitably affect overall performance. It's possible the classifier might exhibit false positives on challenging examples, such as perturbed queries related to the knowledge intended to be forgotten.
> > Given that the forget data for training is quite small, for example, 400 samples for ToFU-10%, is there a risk that the classifier overfits on the training forget data? Can it generalize to other paraphrased versions of the forget data?
>
> We conducted an additional study using multiple types of challenging queries either written by humans, including rephrasing, adversarial, adding irrelevant context, jailbreak, and including only keywords. Below are the false positive/negative rates (in percentage) of the classifier on the perturbed queries:
>
> |Perturbation type|False Positive Rate (%)|False Negative Rate (%)|
> |--------------------------|-----------------------|-----------------------|
> |None|0.0|0.0|
> |Rephrased|0.14|1.5|
> |Adversarial|0.2|1.5|
> |Withirrelevantcontext|0.17|0.5|
> |Withjailbreak-likesyntax|1.65|7.52|
> |Keywordsandshortphrases|0.28|2.51|
>
> Based on these statistics, we believe that our classifiers remain robust under these common perturbations, even without training on these types of data. We continue to explore if we could improve the robustness of our classifiers further. We construct another set of perturbed prompts, all synthetically generated by Llama-3-70B-Instruct, and have no overlap with the previous perturbed evaluation set, and has no knowledge about the perturbation types. We use this set of prompts to train our classifier, and observe that both the false positive rate and the false negative rate can be further improved.
>
> |Perturbation type|False Positive Rate (%)|False Negative Rate (%)|
> |--------------------------|-----------------------|-----------------------|
> |None|0.0|0.0|
> |Rephrased|0.03|0.5|
> |Adversarial|0.06|0.75|
> |With irrelevant context|0.0|0.25|
> |With jailbreak-like syntax|0.08|1.75|
> |Keywords/short phrases|0.0|0.5|
>
> We also perform additional training on the WMDP and copyrighted content classifiers and the result still holds:
>
> WMDP
>
> |Perturbation type|False Positive Rate (%)|False Negative Rate (%)|
> |--------------------------|-----------------------|-----------------------|
> |None|0.00|0.00|
> |Rephrased|0.01|1.00|
> |Adversarial|0.1|1.1|
> |With relevant context|0.00|0.83|
> |With jailbreak-like syntax|0.07|3.50|
> |Keywords/short phrases|0.00|1.03|
>
> Copyrighted content
>
> |Perturbation type|False Positive Rate (%)|False Negative Rate (%)|
> |--------------------------|-----------------------|-----------------------|
> |None|0.0|0.0|
> |Rephrased|0.01|0.2|
> |Adversarial|0.03|1.81|
> |With relevant context|0.00|0.37|
> |With jailbreak-like syntax|0.09|2.63|
> |Keywords/short phrases|0.0|0.18|

---

> > ### Comment · Reviewer_jeTk · 2024-08-09
> >
> > Thanks for the reply, I have read the rebuttal. The selection of surrogate metric looks reasonable to me, but it seems a bit heuristic to a specific task or dataset. Also, the performance is highly related to the additional classifier, which might be challenged by more complicated unlearning scenarios. Nonetheless, I will increase my rating due to the simplicity and performance of the proposed method.

---

> ### Author Response · Authors · 2024-08-10
>
> Thank you for your feedback on our rebuttal.
>
> Regarding the surrogate metric, we agree that in the previous experimental setup, we select the $\hat{v}_r$ in different ways for each task. Here, we just performed **an additional experiment using dataset/task-agnostic selection of $\hat{v}_r$**. Specifically, we use Llama-3-70B-Instruct to generate 100 synthetic responses of either 1) stating "I do not know the answer to the question," or 2) refusal (e.g., "I cannot answer the question"). These responses are all independent of the datasets and tasks considered. Then, we also use the LLM subjected to unlearning to generate its original response (before unlearning). We use the four text similarity metrics used in the copyrighted content unlearning task to measure the difference between 1) the original responses and 2) the synthetic IDK/refusal answers, and minimize this difference for all three tasks. In essence, we want to **push the model output toward IDK or refusal using the zeroth order objective toward minimal textual similarity between the model output and template responses, regardless of the task**.
>
> Due to the time constraint, we only conducted this experiment on Llama-2-7B-Chat, which is a model used in all three tasks.
>
> | Task | Task-dependent selection of $\hat{v}_r$ | Task-agnostic selection of $\hat{v}_r$ |
> | --- | --- | --- |
> | TOFU (forget quality $\uparrow$) | 0.9674 | 0.9188 |
> | WMDP Chemistry (accuracy $\downarrow$) | 26.6 | 25.8 |
> | BBC News (ASG $\downarrow$) | 11.8 | 11.8 |
>
> Note that the score in row 3 does not change because we used the same way to select $\hat{v}_r$ for copyrighted content tasks. Here, we show that **task-agnostic selection still maintains the effectiveness of unlearning**. We will add this particular experiment and result and conduct it on more models as time allows in our revised paper.
>
> Regarding the classifier, I agree that simple classifiers can be challenged in more complicated scenarios. In the future, we will experiment with incorporating external moderation techniques like LlamaGuard or ShieldGemma, which are trained specifically to guard relevant content and are much more robust against jailbreak prompts.

---

### Official Review · Reviewer_WoTW · 2024-07-13

**Soundness:** 3
**Presentation:** 3
**Contribution:** 3
**Rating:** 7
**Confidence:** 4

**Summary:**

The paper proposes Embedding-COrrupted (ECO) Prompts, a lightweight framework for unlearning in large language models (LLMs). It reduces computational inefficiency by using a prompt classifier to identify and corrupt prompts that should be forgotten. In their experiments, this approach, involving zeroth order optimization of prompt embeddings, achieves effective unlearning with minimal side effects and scales efficiently across various LLM sizes without additional costs.

**Strengths:**

- The paper is well-written. The motivation is clear and timely. They cover necessary background, and related work so the readers can easily follow what they are reading.
- The proposed method is lightweight, which can facilitate easy implementation for real world language models.
- The results over comprehensive experiments across multiple datasets and models demonstrate the advantages of the proposed method.

**Weaknesses:**

- Can you describe knowledge entanglement in detail or define it formally? How far is it from structural unlearning, where the removal of one entity might impact knowledge relevant to its neighbours?

[1]: PISTOL: Dataset Compilation Pipeline for Structural Unlearning of LLMs

**Questions:**

see weakness

**Limitations:**

limitations are discussed in appendix

---

> ### Author Rebuttal · Authors · 2024-08-06
>
> We appreciate your recognition of our efforts and your thoughtful comments. Below we address the question about knowledge entanglement.
>
> > Can you describe knowledge entanglement in detail or define it formally? How far is it from structural unlearning, where the removal of one entity might impact knowledge relevant to its neighbours?
>
> In the LLM unlearning domain, we have only seen explicit discussions of knowledge entanglement from two prior works [1, 2]. The TOFU paper describes this as "unexpected forgetting" of other things when one wants to unlearn a specific piece of information, and they also relate this to catastrophic forgetting in continual learning [3]. The authors observed this phenomenon across different models and multiple gradient-based unlearning algorithms. Another study [2] also observed unintended damage in closely related domains when they unlearned information about Harry Potter. However, it seems that a formal definition of knowledge entanglement is lacking. In our paper, we consider knowledge entanglement as the performance drop on datasets that are not relevant to the unlearning target (i.e., law/jurisprudence, math/physics, economics/econometrics, etc.) and show that our method is typically not affected by such problems and outperforms other baselines.
>
> The PISTOL paper [4] is closely related to the concept of knowledge entanglement, where unlearning leads to unintended forgetting of relevant concepts in the knowledge graph. Since the datasets are synthesized from knowledge graphs, investigating structural unlearning and knowledge entanglement should be easier. The graph serves as the ground truth, allowing access to the forgetting of relations or nodes.
>
> While a precise definition of knowledge entanglement remains elusive, we will conduct a more thorough literature review on the topic and include a section discussing the relevant research. We will also cite the PISTOL paper and discuss structural unlearning in the revised version of the paper.
>
> [1] Maini, P., Feng, Z., Schwarzschild, A., Lipton, Z. C., & Kolter, J. Z. (2024). Tofu: A task of fictitious unlearning for llms. arXiv preprint arXiv:2401.06121.
>
> [2] Lynch, A., Guo, P., Ewart, A., Casper, S., & Hadfield-Menell, D. (2024). Eight methods to evaluate robust unlearning in llms. arXiv preprint arXiv:2402.16835.
>
> [3] McCloskey, M., & Cohen, N. J. (1989). Catastrophic interference in connectionist networks: The sequential learning problem. In Psychology of learning and motivation (Vol. 24, pp. 109-165). Academic Press.
>
> [4] Qiu, X., Shen, W. F., Chen, Y., Cancedda, N., Stenetorp, P., & Lane, N. D. (2024). PISTOL: Dataset Compilation Pipeline for Structural Unlearning of LLMs. arXiv preprint arXiv:2406.16810.

---

> > ### Comment · Reviewer_WoTW · 2024-08-08
> > **thank you**
> >
> > Thank you for your response. It would be great if you can incorporate the discussion in the paper as both terms ("knowledge", "entanglement" in the word "knowledge entanglement" are ambiguous. If you can better contextualize the challenge of "knowledge entanglement" in the literature of unlearning, catastrophic interference and potentially "superposition" and position your method among other relevant methods, the paper would be more useful to the community.
> >
> > One follow-up question, do you think knowledge entanglement is caused by superpositioning https://transformer-circuits.pub/2022/toy_model/

---

> > > ### Author Response · Authors · 2024-08-08
> > > **Response to Follow-Up Question**
> > >
> > > Thank you for your suggestions on potential literature we may have overlooked. We will consider a better way to motivate and contextualize the problem in our revision.
> > >
> > > Although we are not experts on superpositioning, we believe that superposition could be a contributing factor. Most current unlearning methods are gradient-based and modify weights in a manner that is likely not feature-aware. This lack of awareness means that entangled knowledge within superposed neurons can be inadvertently damaged during the unlearning process, leading to unintended forgetting. A careful algorithmic design focusing on disentangling features is essential to mitigate this issue.
> > >
> > > An interesting direction for future research would be to study how features are entangled in the network and disentangle them before unlearning, providing a more granular approach to knowledge unlearning. Our approach trains a classifier to explicitly modeling the disentanglement, thus avoiding the problem from the start.
> > >
> > > Besides, it would also be interesting to investigate whether features are "less entangled" when the model sizes increases while the number of features stays constant (?) (or does the number of learnable features also grow?).

---

> ### Comment · Reviewer_WoTW · 2024-08-10
>
> Thank you for your response. I appreciate the detailed explanation and the interesting directions you've proposed for future research. Given your commitment to incorporating relevant discussions on these points in your revision, I have decided to increase the scores. I look forward to seeing how these insights will enhance the final version of the paper

---

> ### Author Response · Authors · 2024-08-10
> **Thank You**
>
> Thank you once again for your thoughtful feedback and for raising the scores! We feel fortunate to have such a knowledgeable and insightful reviewer.
>
> Your comments and suggestions, especially 1) on the literature we have overlooked and 2) on how to provide a clearer presentation of the key concepts, have been invaluable, and we are eager to incorporate these insights to further enhance the final version of our paper.

---

### Author Response · Authors · 2024-08-06
**Summary of Reivews**

We would like to first thank all ACs and reviewers for handling our submission. We value their acknowledgment, helpful critiques, and insightful inquiries about our study. Given the heavy workload of reviewing this year, we quickly summarize the reviews we received.

We received 4 reviews in total, with initial ratings of 6, 4, 4, and 6. All reviewers acknowledged various aspects of our proposed method/paper:

- Lightweight and efficient method (Reviewer WoTW, jeTk)
- Strong performance compared to baseline methods (Reviewer jeTk, qM3a)
- Comprehensive experiments across multiple datasets and models (Reviewer WoTW, UAgG, qM3a)
- Well-written and easy to follow (Reviewer WoTW, UAgG, qM3a)
- Clear and timely motivation (Reviewer WoTW, UAgG)
- Extensive literature review (Reviewer qM3a)

Meanwhile, the reviewers raised several concerns and questions about our work:

- One concern shared by Reviewer jeTk and Reviewer qM3a is **the robustness of our classifier on perturbed prompts** which could potentially bypass the guardrail. While our original goal is to safeguard i.i.d. prompts and achieve unlearning, we acknowledge that prompt perturbation poses a risk, even with minimal effort from attackers. Therefore, we 1) conduct additional evaluation on the classifiers against multiple types of perturbed queries and 2) further train them to detect perturbed prompts.
    1. We show that the classifiers, while not being trained to be robsut, remain effective against perturbed prompts.
    2. By incorporating purely synthetically generated perturbed prompts, we show that the false negative rate of our classifiers regarding the unlearning target can be further reduced to at most 4%.

- One question shared by Reviewer jeTk, UAgG, and qM3a is **the role of the surrogate retain metric value, and how it is obtained**. In equation (8), our goal is to optimize toward the performance of the retained model on the relevant task. However, a retained model is almost always infeasible to obtain in practice, which is why we choose to use a “surrogate” score to approximate how a retained model would perform. We achieve this by either knowing this from the problem context (for WMDP we assume a retained model has a classification accuracy of random-guessing level), or by simulating the responses of a retained model by asking humans to write example responses for a subset of the prompts (100 of them) in our training data. These responses resemble what we believe a retained model would produce. Based on these pseudo responses, we calculate the surrogate retain metric value under the relevant metrics to approximate the real retain metric value.

In our rebuttal, we have clarified the key points and addressed the concerns raised by Reviewers WoTW, jeTk, and UAgG. As a result, they have increased their scores from 6/4/4 to 7/5/5, respectively.

---

### Decision · Program_Chairs · 2024-09-25

**Decision:**

Accept (poster)

**Comment:**

This is a nicely-written paper on LLM unlearning. The idea is to sidestep standard approaches and to just use what amounts to guardrailing. This is a two-step technique: first, determine if the model is going to produce an output that is meant for the model to not give, the second, add noise to token embeddings in such a way that this undesired output is not produced.

While the paper does not produce something totally revolutionary, the approach is well-crafted, the results are solid, and there's a bunch of nice pieces to explore for future work. The reviewers were generally positive; several raised their score after the authors provided detailed explanations to concerns coming up during rebuttal.